# Epithelial–Mesenchymal Transition Signaling and Prostate Cancer Stem Cells: Emerging Biomarkers and Opportunities for Precision Therapeutics

**DOI:** 10.3390/genes12121900

**Published:** 2021-11-27

**Authors:** Luiz Paulo Chaves, Camila Morais Melo, Fabiano Pinto Saggioro, Rodolfo Borges dos Reis, Jeremy Andrew Squire

**Affiliations:** 1Department of Genetics, Medicine School of Ribeirão Preto, University of São Paulo, Ribeirão Preto 14048-900, SP, Brazil; luizpaulocds@usp.br (L.P.C.); camila.morais.melo@gmail.com (C.M.M.); 2Pathology Department, Medicine School of Ribeirão Preto, University of São Paulo, Ribeirão Preto 14048-900, SP, Brazil; fsaggioro@terra.com.br; 3Division of Urology, Department of Surgery and Anatomy, Medicine School of Ribeirão Preto, University of São Paulo, Ribeirão Preto 14048-900, SP, Brazil; rodolforeis@fmrp.usp.br; 4Department of Pathology and Molecular Medicine, Queen’s University, Kingston, ON K7L 3N6, Canada

**Keywords:** castrate-resistant prostate cancer, immunotherapy, immune evasion, tumor microenvironment, oncogenes, tumor suppressor genes, mouse models of cancer, epigenomics, plasticity, chromatin modification

## Abstract

Prostate cancers may reactivate a latent embryonic program called the epithelial–mesenchymal transition (EMT) during the development of metastatic disease. Through EMT, tumors can develop a mesenchymal phenotype similar to cancer stem cell traits that contributes to metastasis and variation in therapeutic responses. Some of the recurrent somatic mutations of prostate cancer affect EMT driver genes and effector transcription factors that induce the chromatin- and androgen-dependent epigenetic alterations that characterize castrate-resistant prostate cancer (CRPC). EMT regulators in prostate cancer comprise transcription factors (*SNAI1/2*, *ZEB1*, *TWIST1*, and ETS), tumor suppressor genes (*RB1*, *PTEN*, and *TP53*), and post-transcriptional regulators (miRNAs) that under the selective pressures of antiandrogen therapy can develop an androgen-independent metastatic phenotype. In prostate cancer mouse models of EMT, Slug expression, as well as WNT/β-Catenin and notch signaling pathways, have been shown to increase stemness potential. Recent single-cell transcriptomic studies also suggest that the stemness phenotype of advanced prostate cancer may be related to EMT. Other evidence correlates EMT and stemness with immune evasion, for example, activation of the polycomb repressor complex I, promoting EMT and stemness and cytokine secretion through *RB1*, *TP53*, and *PRC1*. These findings are helping clinical trials in CRPC that seek to understand how drugs and biomarkers related to the acquisition of EMT can improve drug response.

## 1. Prostate Cancer Progression

Prostate cancer (PCa) is the second most common cancer in men and the fifth cause of cancer-related deaths worldwide [1,2]. It is expected that this disease will account for 1 in 5 new cases of cancer in American men in 2020 [2]. For men diagnosed with advanced and metastatic PCa, the 5-year survival rate is 30% [3]. Androgen deprivation monotherapy (ADT) is no longer the standard of care for men with metastatic hormone-sensitive PCa (mHSPC); today, ADT should be combined with second-generation antiandrogens (enzalutamide, apalutamide, darolutamide, or abiraterone acetate) or chemotherapy (docetaxel), although ADT is a unique therapy that should indefinitely continue in all stages of metastatic disease [4]. Resistance to chemotherapy and hormonal therapy in the metastatic setting remains a significant cause of morbidity and mortality [5].

There have been great improvements in understanding the landscape of genomic alterations and somatic mutations in PCa in recent years. Key somatic mutations in PCa include fusions of *TMPRSS2* with *ETS* family genes, amplification of the *MYC* oncogene, deletion and/or mutation of *PTEN*, *RB1*, and *TP53* in advanced disease, together with amplification and/or mutation of the androgen receptor (AR) [3]. At the pathway level, PI3K, DNA repair, WNT/β-catenin, and epigenetic regulators are more frequently altered in metastatic samples [6,7,8].

The most common somatic rearrangements in PCa are gene fusions that involve members of the *ETS* family of transcription factors such as *ERG*, *ETV1*, and *ETV4*, which have important roles in embryonic development and cell proliferation. The most frequent of these fusions is between the AR-dependent promoter of the *TMPRSS2* gene onto the coding region of the *ERG* gene. This gene fusion has been found in fifty percent of primary PCa, leading to AR-dependent overexpression of *ERG* in these tumors [9].

Second-generation antiandrogens such as abiraterone and enzalutamide have been highly effective in controlling AR signaling activity, but unfortunately, most metastatic PCa develop antiandrogen resistance [10]. In many cases, CRPCs exhibit ‘‘androgen-indifferent’’ phenotypes, containing regions of tumor that lack AR expression. These histologically AR-negative tumors may also display lineage plasticity (see Table 1), as in the case of neuroendocrine prostate cancer (NEPC) (see Table 1). Other tumors lack both AR expression and neuroendocrine differentiation, and they are called ‘‘double-negative’’ PCa (DNPC), identified in approximately 25% of metastatic CRPC patients [11]. AR deregulation in more advanced PCa is associated with chromatin reprogramming of AR cistrome and epigenetic alterations, which can trigger neuroendocrine differentiation. As we discuss below, it is thought that the various acquired genomic rearrangements and chromatin changes that characterize CRPC overcome the effects of hormonal therapy by reactivation of prostate developmental stem-cell-like gene expression [12].

In this review, we summarize our current understanding regarding the role of EMT in PCa with a specific focus on the relationship of somatic mutations of PCa on CSC biology and the immune microenvironment. These findings will be related to PCa translational research that implicates the function of *TGF-β*, *ZEB1*, *TMPRSS2-ERG* fusion, and *PTEN* in driving EMT and activating pathways associated with therapeutic resistance.

Accumulating evidence suggests that some of the common somatic mutations of PCa not only impact classical hallmarks of cancer pathways [13], but also elicit cellular changes in the tumor microenvironment (TME), allowing cancer cells to avoid immunosurveillance [14]. One of the molecular processes associated with aggressive cancers that is also known to impact the TME is EMT [15].

## 2. Epithelial to Mesenchymal Transition (EMT) and Cancer Progression

When a cancer undergoes EMT, tumor cells lose their epithelial characteristics and acquire mesenchymal features (Figure 1). EMT is often followed by the reverse process of MET [16]. EMT has been associated with various tumor functions, including tumor initiation, malignant progression, tumor stemness, tumor cell migration, intravasion to the blood, metastasis, and resistance to therapy. The cellular changes that accompany EMT include enhanced motility and loss of cell–cell adhesion that are associated with a more aggressive cellular phenotype [17]. EMT has been classically characterized by loss of the epithelial markers E-cadherin and β-catenin, accompanied by an increase in cell migration and invasiveness due to activation of a specific transcriptional program. The hallmark mechanism of EMT in tumor progression is loss of cell–cell adhesion [18]. Transcriptional regulators, such as *TWIST*, *SNAI1*, *SNAI2*, *ZEB1*, and *ZEB2*, repress E-cadherin expression, while others promote the expression of mesenchymal differentiation markers, such as N- and/or R-cadherin and vimentin, as well as the expression of cellular matrix and focal adhesion proteins [19]. Some of these EMT transcription factors also function as biomarkers of less favorable disease course [20]. EMT alterations of cadherin also contribute to changes in the stroma-oriented cellular adhesion profile, with increased tumor cell motility and invasive properties associated with metastatic disease. Progression of PCa to mCRPC may be associated with the expression of cellular adhesion molecules, which mediate aberrant interactions between glandular epithelial cells and the extracellular matrix in the TME. There is a strong association between reduced expression of E-cadherin and membranous β-catenin and progression in various types of carcinomas, including PCa, which could be used as a prognostic biomarker [21]. Prostate tumors are also able to interact with supporting extracellular matrix and stromal elements to establish a pro-tumorigenic immunosuppressive TME [14,22].

Tumors undergoing EMT may express intermediate morphological, transcriptional, and epigenetic features, that range across the spectrum of epithelial and mesenchymal markers [23]. These different phenotypic states, often referred to as quasi-mesenchymal intermediates, that vary between epithelial and a complete mesenchymal expression, have been referred to in the literature as partial, incomplete, or hybrid EMT states [24]. It is thought that the plasticity associated with EMT is likely a reflection of these intermediate expression states [25].

Cellular plasticity (discussed in Section 5) is not only a characteristic of EMT but is also strongly associated with the cancer stem cell (CSC) behavior [26]. Both normal stem cells and CSCs have shown associations with EMT in various tissue types [27], and evidence favors an important role for CSCs in cancer metastasis [24]. The CSC hypothesis is an emerging model that helps explain many of the molecular characteristics of malignancy as well as the propensity of human tumors to relapse, metastasize, and acquire resistance to conventional therapies. CSCs have been identified in murine PCa model systems [28], PCa cell lines [29], and isolated from patient tumors [30] using stem cell-specific surface markers and various in vitro enrichment assays, such as PCa organoid culture [31].

These two important cellular programs of PCa progression—the CSC phenotype and EMT—are strongly interconnected [32].

## 3. Mouse Model and In Vitro Studies of EMT in PCa

Diverse mouse and in vitro studies using various experimental EMT treatments of carcinoma cells can induce expression of mesenchymal markers, leading to acquisition of a mesenchymal phenotype [33,34]. In vivo metastasis models have been particularly helpful for investigating interactions between tumor cells with tumor-associated stromal components and the TME [35]. Organoid culturing methods have also provided insights about induction of differentiation in vitro and have provided excellent model systems to show how signals inducing EMT may also influence immune anti-tumor responses and therapy resistance [24].

Ruscetti et al. used mouse models to characterize the in vivo role of EMT and investigate how mesenchymal plasticity (see Table 1) might influence metastatic potential in PCa [36]. The authors developed an elegant in vivo tracking system that allowed them to investigate the capacity for tumor initiation of mesenchymal-like EMT intermediate tumor cells compared to mesenchymal cells that have fully completed EMT. They showed that both the intermediate mesenchymal and the fully expressing EMT mesenchymal tumor cells were able to initiate primary tumors. However, only the mesenchymal-like intermediate tumor cells could persist in circulation and survive in the lung following intravenous injection. The mesenchymal-like intermediate tumor cells appear to exhibit characteristics of quiescent stem cells, whereas the fully expressing EMT tumor cells exhibit characteristics of proliferating progenitor cells that have lost stemness features. The authors point out that since mesenchymal-like intermediate tumors usually localize to the stem cell niche in the proximal region of the prostate gland, it is likely that they are able to maintain their dormant quiescent state by growth factors provided by the surrounding microenvironment at metastatic sites [37].

A study using xenograft growth of the PC3 prostate cancer cell line showed that PCa expression of IL6 activates a specific type of cancer-associated fibroblast which induces EMT, invasiveness, and stemness [38]. FGF signals appear to be important in PCa progression since this growth factor is usually overexpressed in advanced PCa, such as DNPCs [11]. Expression of the growth factor receptor (*Fgfr1*) also seems to have EMT as an effector of PCa progression via *Sox9* and Wnt signaling [39]. This suggests the potential role of *IL6* and FGF as prognostic biomarkers.

Prostate tumors interact closely with the TME and stromal elements adjacent to tumor cells, which creates an immunosuppressive TME [14,22]. An early study on this topic highlights the role of cancer-associated fibroblast (CAFs) in the TME of PCa. These TME cells interacted with polarized M2 macrophages to stimulate the CAFs toward cooperative cellular activities with stemness traits and an EMT phenotype that increased PCa tumor cell motility and metastatic spread [40].

Research using a PBCre4:*Pten^f/f^* mice, a *Pten* loss-induced prostate adenocarcinoma model, showed that *Rb1* loss is an important driver of lineage plasticity, which is characterized by enhanced EMT and stemness potentials (see Section 5). Transcriptomic profiling suggests that this phenotype, in both humans and mice, has the epigenetic reprograming factors *Sox2* and *Ezh2* as effectors [41]. About 50% of NEPCs inactivate both the *Rb1* and *Tp53* tumor suppressor genes. Two studies have shown that loss of *Tp53* and *Rb1* is strongly associated with acquisition of lineage plasticity and epigenetic alterations, leading to resistance to androgen deprivation therapy [41,42]. Mu et al. showed that loss of both *Rb1* and *Tp53* induces a *Sox2*-mediated cellular plasticity, as indicated by decreased expression of luminal epithelial cell markers and increased expression of basal and neuroendocrine markers [42].

Contactin1 (*Cntn**-1*) is a cell adhesion neuronal membrane glycoprotein belonging to the immunoglobulin superfamily. The protein has been shown to utilize EMT-dependent promotion for enhanced cell invasion, migration, and metastasis in several different types of carcinoma [43]. Downregulation of *Cntn-1* resulted in decreased activity of the PI3K/Akt signaling and docetaxel resistance in PCa cells lines and xenografts [44]. Since *Pten* loss and PI3K/AKT dysregulation are strongly associated with advanced PCa and CRPC, this preclinical finding draws attention to the potential collaborative role between EMT and common PCa driver mutations.

*Pten* loss has also been shown to promote EMT, and recent work suggests that *Pten* restoration in breast cancer models inhibits EMT and stemness CSCs activity via Abi1 (Abelson interactor 1) downregulation [45]. Abi1 is an adapter protein that has been implicated in actin cytoskeletal remodeling and intercellular adhesion. Recently, Abi1 was shown to control progression and epithelial plasticity in PCa through regulation of the EMT-WNT pathway [45]. Abi1 was found to control EMT downstream of the non-canonical WNT receptor *Fzd2* and upstream of the active *FYN-STAT3* axis [45]. TGF-β1, another known EMT driver, was shown to do so via alternative splicing of CD44 to an isoform that promotes progression through migration, invasion, and tumor initiation [46]. The NOTCH signaling is a critical pathway involved in EMT and metastasis, and a recent investigation has indicated that it is an important player in stem-like basal cells through activation of the estrogen receptor [47].

Lineage plasticity in metastatic CRPC is likely to be mediated by epigenetic reprogramming, at least in part through histone modifications performed by polycomb repression. The polycomb repressive complex is an important player in EMT regulation. Previous work has shown that extracellular Hsp90 has the epigenetic function of reversing the polycomb function to drive tumor growth and invasion through EZH2, the enzymatic subunit of Polycomb Repressive Complex 2 (PRC2) [48]. As discussed above, *EZH2* is believed to play a key role in promoting epigenetic changes, leading to neuroendocrine differentiation. This is affected by methylating histone H3 lysine 27 at the promoter of target genes [38]. Interestingly, polycomb regulation is also involved in mediating stem cell functions. The canonical PRC1 subunit Bmi1 has been implicated as a cellular marker of castrate-resistant luminal stem cells in PCa [49]. This finding is consistent with other studies implicating a regulatory role for PRC1 in CRPC. Inactivation of PRC1 in the normal prostatic epithelium causes cell-autonomous expression of Ccl2 and other inflammatory cytokines [50]. Su et al. found that PRC1 drives metastasis of DNPC tumor subtypes by regulating Ccl2 expression. Ccl2 in the TME of PCa signals self-renewal and recruitment of tumor-associated macrophages (TAMs), myeloid-derived suppressor cells (MDSCs), and T-regulatory cells (Tregs), which lead to immunosuppression and increased angiogenesis. Importantly, PRC1 inhibition combined with immune checkpoint blockade reversed these processes and suppressed metastasis in PCa mouse models [51]. Along with further investigations, these findings could support the use of CCL2 expression in precision medicine as a predictive biomarker for immunotherapy response.

The genomic background is an important player in defining the TME and immunological landscape. A study from Bezzi et al. showed that different genetically engineered mouse models (GEMMs) carrying homozygous *Pten* loss led to marked differences in the immune composition of the TME in induced tumors. Loss of the *Zbtb7a* gene in addition to *Pten* loss induced higher expression of Cxcl5, while loss of *Tp53* in addition to *Pten* loss induced higher expression of Clcl17, which correlates to myeloid cells’ attraction through different mechanisms [52].

Classical studies of the normal prostatic epithelium in rodents show that the gland comprises basal cells, secretory luminal cells, and rare neuroendocrine cells surrounded by stroma and vasculature [53]. Using various types of in vitro organoid-forming assays [31], combined with stem cell enrichment assays, the existence of prostate stem/progenitor cells has been demonstrated in both basal and luminal prostate epithelial lineages [28,29,30,54,55,56].

One of the first studies of murine PCa suggested that tumors may arise from CSCs that undergo *Pten/Akt* signaling dysregulation and produce a heterogeneous population of cells, expressing luminal, basal, and intermediate cells [57]. Wang et al. identified a CSC-like subpopulation expressing ZEB1 in prostatic basal stem cells [58]. Emerging evidence suggests that ZEB1 drives the induction of EMT with activation of a subpopulation that have stem cell traits, immune evasion, and epigenomic reprogramming (see Section 5) [59]. The authors used single-cell RNA-seq analysis to show that the Zeb1 subpopulation had mesenchymal cellular features and higher expression levels of EMT inducers such as Snai1, Zeb2, Prrx1, and Prrx2 [58]. Another study showed that basal cells preferentially express genes that confer intrinsic stem-like and neurogenic properties. Of clinical relevance, the basal cell gene expression profile was found to be enriched in advanced, anaplastic, castration-resistant and metastatic PCa [60]. Similarly, Alumkal et al. demonstrated that a basal lineage, neurogenic/stemness program was activated in ADT non-responders to AR inhibition therapy, while a luminal lineage program was more active in responders (discussed in Section 5) [61].

Collectively, these analyses suggest that rare subpopulations of basal cells have the potential to function as ADT-resistant CSCs in PCa. However, the precise origin of CSCs in advanced PCa remains inconclusive as another single-cell RNA-seq study indicated that rare luminal cell types also had stem-like regenerative properties [54]. The increasing use of single-cell RNA-seq analysis of CSCs from tumors at different stages of the EMT spectrum will provide more detailed transcriptomic signatures that will help to elucidate which lineages contribute to CSCs and are associated with PCa progression, drug resistance, and metastatic disease.

## 4. Transcriptional Signatures and EMT Pathways of Progression

The development of high-throughput omics technologies allowed a better understanding of how different mechanisms and processes work at the molecular level. While phenotypic evaluation of EMT has more general definitions [15] (see definitions in Table 1), transcriptomics and genomics analysis is providing specific gene signatures associated with the various intermediate-mesenchymal states (discussed in this section). EMT transcriptional ‘drivers’ and ‘effector’ genes may both function as biomarkers along the spectrum of EMT [16] (see Figure 2). Since these genetic signatures may be related to cancer stem cell biology, and clinical outcomes in PCa (as discussed in the previous section [36]), this area of EMT research is particularly important.

Byers et al. developed an EMT signature for non-small-cell lung cancer, integrating transcriptomics, genomics, and drug response analysis in vitro and in vivo. This preclinical study showed that this EMT signature could classify tumors as epithelial-like or mesenchymal-like, and that the mesenchymal-like intermediate states were associated with poor outcomes, such as drug resistance. Their analysis also identified the receptor tyrosine kinase Axl as a novel biomarker of EMT in this type of cancer and as a potential therapeutic target, along with EGFR inhibition [63]. Computational modeling has been used to investigate the interconnection between EMT, stemness, and Notch signaling. The authors analyzed CSC gene expression profiles from different types of cancer and showed that stemness properties can be found in any of the different phenotypes along the EMT spectrum. Their data suggest that Notch-jagged signaling could underlie gene expression relationships between stemness and EMT [64]. Their model also draws attention to the potential anti-cancer role of the diabetes drug metformin, which shows promise for therapeutic inhibition through the Notch-Jagged pathway in tumors with EMT (discussed in Section 7).

Other groups have used similar mathematical models of gene expression data to devise diverse classification metrics of EMT activities. George et al. developed a predictive model to score the degree to which different cancer cell lines exhibit intermediate EMT features. They showed that EMT status based on their model was associated with survival in a tissue- and subtype-specific manner [65]. Another group developed a molecular signature for non-small-cell lung cancer (NSCLC) with 16 genes and showed that EMT was associated with immune cell exclusion and expression of immunosuppressive cytokines [66]. In head and neck squamous cell carcinoma, an 82-gene molecular signature for EMT was functionally validated in vitro and could predict for poor outcome and immune evasion [67].

The increasing interest in cancer immunotherapy has meant that analysis of correlations between EMT, stemness, and immune response has become an area for investigation. A study using genetically engineered mouse models of PCa showed that the immunological landscape of the TME can be genetically driven [52]. Thorsson et al. molecularly characterized six distinct immune subtypes (called C1–C6) across 33 cancer types in TCGA [68]. The immune subtypes were characterized by dominance of either macrophage or lymphocyte signatures, T-helper phenotype, the extent of intra-tumoral heterogeneity, and by proliferative activity. Another deconvolution study of bulk RNA-seq public domain cancer data established scoring metrics across the EMT plasticity spectrum. They compared their metrics to the C1–C6 immune subtypes of Thorsson et al. Their predictive analysis showed that different EMT scores were associated with specific immune subtypes. According to the authors, a more epithelial-like score is related to wound healing and IFN-γ subtypes (C1 and C2, respectively), while the most mesenchymal-like score was related to the quiescence subtype (C5), which would be consistent with an increased mesenchymal score also having stemness features [69]. This was the first step toward a prolific use of molecular signatures associated with EMT in precision therapy, since different molecular subtypes could be efficient prognostic and predictive biomarkers to stratify patient populations.

Transcriptional investigation of PCa showed that epithelial plasticity is related to poor clinical outcome, but that the MET reverting transition signature also has important properties [70,71]. Stylianou et al. showed that the EMT molecular profile was enriched in patients recently subjected to ADT. Treatment of hormone-sensitive PCa with antiandrogen therapies imposes a strong negative selection pressure on tumors. Thus, ADT may select for chromatin remodeling, epigenetic plasticity, and widescale transcriptional changes in surviving ADT-resistant tumor cells (discussed in Section 5). The authors also showed that the epithelial–mesenchymal plasticity signature, which incorporates both EMT and MET, was associated with metastasis and predicted a poor patient prognosis. Additionally, they showed that in their PCa model, EMT was mainly driven by SNAI1. Snail proteins are zinc finger transcriptional repressors controlling EMT that are also involved in the acquisition of migratory properties to epithelial cells. Inhibition of Snail alone decreased the expression of mesenchymal markers such as the effectors *ZEB1* and vimentin and restored expression of epithelial markers such as CDH1 [72]. *SNAIL* gene deregulation appears to be involved in AR signaling since loss of SNAI2, another Snail protein, was associated with a better response to androgen deprivation therapy [73]. BRD4 is a chromatin regulatory protein that promotes expression of the EMT-related transcription drivers SNAI1 and SNAI2, and its ablation represses TGF-β-mediated EMT. Decreased expression of BRD4 was associated with reduced survival in PCa based on TCGA data [74]. The Wnt pathway is also related to EMT. Recently, the non-canonical Wnt pathway Wnt5a/Fzd2 was discovered to increase EMT markers in PCa, and a signature associated with this pathway was developed to predict PCa aggressiveness [75]. This finding was also consistent with the EMT study (discussed in Section 3) in which Abi1 controlled epithelial plasticity downstream of the non-canonical WNT receptor FZD2 [45].

## 5. Epigenomic Regulation of EMT and Lineage Plasticity in Prostate Cancer

The concept of epigenetic plasticity has been used to describe how chromatin structure and altered chromatin states are associated with epigenetic changes that confer downstream transcriptional alterations to cancer hallmark pathways [76]. Lineage plasticity is of particular clinical relevance in advanced PCa since it is a common mechanism of resistance to ADT (Table 1). These increasingly powerful combinations of antiandrogen drugs impose a strong negative selection pressure on tumors so that altered chromatin states accompanied by epigenetic-driven changes to the transcriptome characterize the surviving ADT-resistant tumor cells. Many of the transcriptionally important genes that drive normal prostatic development and differentiation also function as oncogenes (*ETS*, *MYC*, *FOXA1*) or tumor suppressor genes (*RB1*, *PTEN*, *TP53*) in primary and/or CRPC [77,78]. There is increasing awareness of the importance of finding connections between epigenetic and chromatin changes in ADT-resistant CRPC and the role played by the common PCa recurrent somatic mutation [79]. In Figure 2, we highlight how the frequently mutated genes of PCa, such as *PTEN*, *TP53*, *CDH1*, *TMPRSS2-ERG*, and *RB1*, interact with EMT effectors and drivers involved in epigenetic pathways leading to mCRPC.

Emerging epigenomic data show that prostate-specific genes that are normally responsible for cell motility and invasion during fetal organogenesis are targets for re-expression in mCRPC. These fetal developmental genes are usually inactive in adult prostate and primary tumors. They all appear to be AR-dependent genes that become activated by the transcription factors *FOXA1* and *HOXB13* at loci, and have a DNA binding motif for the EMT effector *ZEB1* [78]. These data are consistent with earlier correlative studies indicating that the expression of *ZEB1* is associated with tumor grade and outcome in PCa, highlighting the importance of EMT biomarkers such as *ZEB1* in predicting aggressive disease [80] (see Figure 2).

During EMT, the DNA methylation landscape pattern is steadily altered. Expression of the EMT effector and driver transcription factor families such as *SNAI*, *ZEB*, and *TWIST* seem to be autoregulated by DNA methylation [81]. Activation of the signaling pathways SMADs, PI3K/Akt, and MAPK/ERK through EMT drivers, such as TGF-β, are related to epigenetic-remodeling. ZEB1 is thought to induce EMT by imposing a cellular plasticity phenotype [59]. ZEB2 expression is also involved in cellular plasticity of embryonic stem cells through controlling cellular methylation [81]. Focal hyper-methylation of CDH1 is induced by ZEB1 expression and interactions of EMT transcription factors with methyltransferases [59,81]. In prostate cancer, altering the DNA methylation pattern has been shown to be essential in the fibroblast-induced EMT and stemness [82].

Lysine-specific histone demethylase 1A (LSD1) is part of the chromatin remodeling complexes involved in the balance of self-renewal and differentiation. LSD1 overexpression was observed in different types of cancer and is inversely correlated with CDH1 expression. Both SNAI and ZEB protein families have been shown to interact with this histone-modifying enzyme. SNAI and ZEB families have also been shown to recruit the transcriptional repressor Mi-2/Nucleosome-Remodeling and the Deacetylase (NuRD) Complex to their target promoters. The SWItch/sucrose non-fermentable (SWI/SNF) complex has been shown to interact with ZEB1 and the Wnt signaling pathway to promote EMT. Lastly, both polycomb repressive complexes 1 and 2 were shown to interact directly and indirectly with EMT transcription factors to promote plasticity and inhibit the tumor suppressor *PTEN* [81].

In a recent single-cell analysis by He et al. based on 14 patients with advanced PCa treated with antiandrogen therapy, the patients were found to co-express multiple AR isoforms, including truncated isoforms. Resistance to therapy was also associated with upregulation of gene signatures for EMT and TGF-β signaling. Additionally, a subset of patient tumors had high expression of dysfunctional cytotoxic CD8+ T cell markers, suggesting that EMT may also impact immune responses in CRPC [83].

## 6. Noncoding RNA Regulation of EMT and Lineage Plasticity

Micro-RNAs can act as epigenetic modulators of gene expression, modifying RNA stability and translation, and in some cases, promoting transcription without modifying the gene sequences [84]. Additionally, micro-RNAs can themselves be regulated by chromatin and epigenetic modifications [85]. The best-known micro-RNAs involved in EMT regulation are the miR-200 family and miR-34. MiR-200 works in a double-negative feedback loop with ZEB1/2 and targets the Wnt/β-catenin pathway. MiR-34 and miR-203 work in an auto-regulated feedback loop with SNAI1 [59,86]. Micro-RNAs can also induce EMT. MiR-21 and miR-31 can suppress TIAM1 to promote *TGF-β* and miR-155 expression, and reduce RhoA to disrupt tight junctions’ formation, and their knockdown was shown to suppress *TGF-β*-mediated EMT [81,87]. The transcription factor Kaiso is a TP53-dependent regulator that directly binds to methylated regions of the DNA. Overexpression of the gene has been shown to downregulate miR-200, possibly through an EGFR-Kaiso signaling axis, to promote *ZEB1* expression and induce EMT in PCa [88].

The EMT regulatory miR-3622a maps to chromosome region 8q21. Since this chromosomal region is subject to recurrent deletion in PCa, and loss is associated with poor prognosis, it has been suggested that deregulation of EMT by miR-3622a may be promoting tumor progression in PCa bearing this deletion [89]. However, it is possible that loss of other genes in this deleted region of chromosomes may also be contributing to PCa outcome.

Long noncoding RNAs (lncRNAs) are also known to regulate EMT effector and driver transcription factors (see Figure 2). ZEB1-AS1 (ZEB1 antisense 1) and ZEB2-AS1 are lncRNAs that are transcribed from a shared bi-directional promoter of *ZEB1*. Both lncRNAs can upregulate *ZEB1* and *ZEB2* expression. HOTAIR, a HOX antisense intergenic RNA with previously described roles in the normal development, seems to be involved in EMT, since it is required for both *SNAI1* and miR-34a repression. MEG3 is a lncRNA that represses both CDH1 and the miR-200 family. Repression is achieved by recruitment of JARID2, EZH2, and histone H3 methylation to the regulatory regions of *CDH1* and miR-200. These alterations lead to increased expression of *TGF-β* [81,90].

Recently, the lncRNAs PVT1 and NORAD have been shown to promote EMT in PCa via mediating miRNA targeting of EMT promoters [91,92]. NORAD was shown to endogenously compete with other targets of miR-30a-5p, downregulating its activity, therefore promoting RAB11A expression and Wnt/β-catenin pathway activation [92]. PVT1 competes with miR-186 targets, which include the EMT effector Twist1 [91].

Epigenetic regulatory mechanisms involve interaction of transcription factors with protein complexes, and noncoding RNAs offer several new therapeutic approaches that target EMT and CSCs [27,93] (Figure 2). Moreover, the lineage plasticity associated with EMT in mCRPC provides an opportunity to determine which genetic driver pathways are associated with drug resistance and immunotherapy failure in advanced disease.

## 7. Clinical Trials

Progress in treating metastatic disease is starting to take advantage of improved understanding of the interconnected signaling pathways that control EMT and stemness pathways during tumor progression. Since EMT is implicated in both cancer metastasis and induction of drug resistance, targeting EMT may have enormous therapeutic value [94]. As discussed above, many EMT drivers are epigenetically regulated, so drugs that influence DNA methylation, histone modification, and plasticity are emerging as potential therapeutic targets for overcoming EMT [95]. In PCa, EMT appears to be an important mediator of acquired resistance to both androgen deprivation and docetaxel therapies [44,61,72].

Driver and effector transcription factors such as Snail, Twist, Zeb, or Stat3 are activated early in EMT [19]. Due to their crucial role in regulating the EMT process, inhibiting their expression may be a way of reversing the EMT process and preventing activation of metastatic pathways. In addition, many crucial EMT signaling pathways, such as TGF-β1, NF-κB, Wnt, Akt, PPAR, and Notch, contribute to the EMT cancer phenotype, so inhibitors of these pathways may also be investigated.

A recent clinical trial (NCT02099864) showed that tumors from CRPC patients who failed to respond to Enzalutamide (a non-steroidal androgen-receptor inhibitor) therapy harbored a transcriptional profile associated to stemness and reduced AR activity in basal lineages [61]. Together, these findings highlight the need to explore the use of novel EMT-targeting agents as adjuvant therapy associated with standard of care therapies in order to retard and inhibit—or maybe even prevent—de novo resistance.

Numerous clinical trials have been developed to target EMT in different types of advanced cancer in an effort to find new biomarkers, minimize progression, and improve therapeutic responses. In advanced PCa, clinical trials have been designed to investigate the use of EMT as new biomarkers for prostate cancer, and some therapeutic agents targeting EMT have reached phase II trials (Table 2).

Most of the clinical trials using EMT try to relate well-established findings to clinically relevant biomarkers in the blood, such as circulating cells and tumor antigens. The purpose is either to capture circulating tumor cells using immunoaffinity mesenchymal attractants or to identify EMT cells and molecules with clinical relevance to predict aggressiveness and therapy response.

Among the therapeutic agents, metformin, a classical anti-diabetic agent, has been studied in cancer prevention and therapy, mainly because of the AMP-kinase activation. In PCa, the antineoplastic effect may be not only through AMPK, but also IGF-1, mTOR, and AR signaling [96,97]. In vitro and observational evidence from preclinical studies also showed a possible action of metformin in preventing stem cells’ self-renewal and proliferation, possibly though Notch targeting [96,98,99].

Adavosertib, a specific WEE1 inhibitor, has also been investigated as an EMT-targeting agent. WEE1 is a negative regulator kinase of CDK1. Loss of cell cycle kinase CDK1 seems to facilitate cells undergoing EMT-mediated escape from immunosurveillance. Specifically inhibiting WEE1 may restore CDK1 expression and therefore re-establish immuno-mediated attack of mesenchymal-like cells [100].

Given the roles of chromatin remodeling as drivers and effectors of cell plasticity, epigenetic-targeting drugs may indirectly regulate EMT and cancer progression. Many specific compounds have been developed that can target the chromatin remodeling effects of EMT. Inhibiting DNA methyltransferases (DNMTs) may avoid hypermethylation of CDH1 and tumor suppressors, while the inhibitory chromatin-modifying effects of EZH2 may represses PRC2 activity. The blocking of histone deacetylases (HDACs) may prevent histone modifications implicated in cancer [101,102]. Targeting epigenetic- and chromatin-related EMT processes in combination with standard therapeutic approaches such as ADT should be a promising strategy for improving disease outcome in PCa. However, a broader understanding of the molecular biology of EMT and its relationship with the established molecular subtypes of advanced PCa is crucial to further implement EMT-targeting strategies of treatment in clinical trials.

## 8. Future Directions and Conclusions

Contemporary EMT research highlights the importance of understanding the interconnections between cell plasticity, stemness, and therapy response. These processes are tightly involved in tumor initiation, invasiveness, migration, metastasis, and interactions with the TME. EMT can be triggered by different drivers and effectors that may function as prognostic biomarkers of metastatic disease. TWIST, SNAIL, and ZEB families of promoters are closely involved in EMT regulation through alterations to the transcriptional landscape and through epigenetics changes such as chromatin modification and noncoding RNA modulation.

During the last ten years, there has been considerable investigation of the role of EMT in cancer progression and therapy resistance [103,104]. In PCa, further investigations are needed to identify the most promising targets to improve treatment of CRPC. *SNAIL* seems to be a crucial driver of EMT in PCa, while Notch and Wnt signaling are more closely involved in both the EMT and cancer stemness phenotypes. Targeting both these pathways may be effective for some types of cancer. Meanwhile, the development of human transcriptional signatures of EMT could be predictive of progression and therapy response. Epigenetic modifications also offer promise, but the downside is that global changes of methylation and histone modification may lead to unexpected alterations in gene expression, with unfortunate adverse events. Targeting specific epigenetic effectors, known to be involved in EMT, such as LSD1, as have been studied in other types of cancers, may also be of great benefit for PCa [102]. Furthermore, a better understanding of how EMT and stemness modulate the immune tumor microenvironment of PCa may provide new insights for novel therapeutic approaches.

Finally, the recent tragic COVID-19 pandemic required the rapid development of novel mRNA vaccines for targeted therapies against SARS-CoV-2. These new mRNA vaccine platforms could be used to encode specific and personalized synthetic peptides that with an appropriate delivery system could modulate key pathways, such as EMT [105]. Technical developments in combination with the growing understanding of the different processes involved in EMT and CSC biology improve the prospects for precision cancer therapy.

## Figures and Tables

**Figure 1 genes-12-01900-f001:**
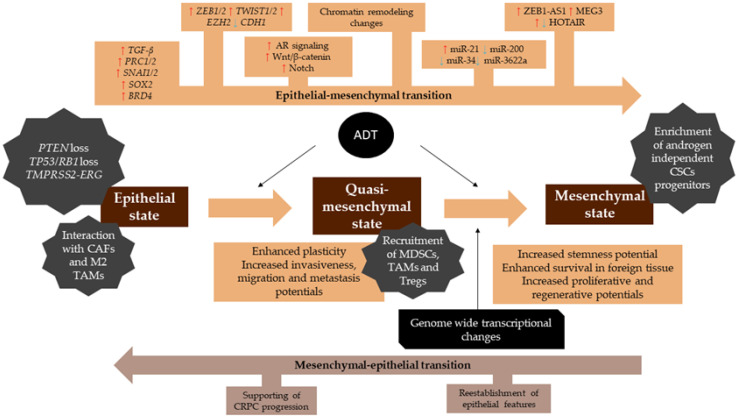
Processes and molecular events involved in EMT in prostate cancer. Tumor cells in an epithelial state gain internal and external stimuli to restore the EMT programming. EMT promotes wide transcriptional changes, illustrated at the top.Red arrows indicate upregulated genes, pathways and noncoding RNAs during EMT, blue arrows indicate the ones that are downregulated during this process. The quasi-mesenchymal state is the state of most of the cells affected by EMT, as shown in the center of the figure. At this stage, cells have enhanced plasticity and increased progression features. Quasi-mesenchymal cells modulate the tumor microenvironment, expressing immune evasion-related cytokines and recruiting immunosuppressive cells, which enhances an immunological evasion phenotype. Cells undergoing EMT confer a selective advantage when patients undergo antiandrogen therapy, enriching androgen-independent CSC progenitors. In the mesenchymal state, cells are usually resistant to therapy and have increased stemness and survival potentials. The mesenchymal–epithelial transition is established to restore epithelial features and support castrate-resistant prostate cancer (CRPC) growth. CAFs—cancer-associated fibroblasts, TAMs—tumor-associated macrophages, MDSCs—myeloid-derived suppressor cells, CSCs—cancer stem cells.

**Figure 2 genes-12-01900-f002:**
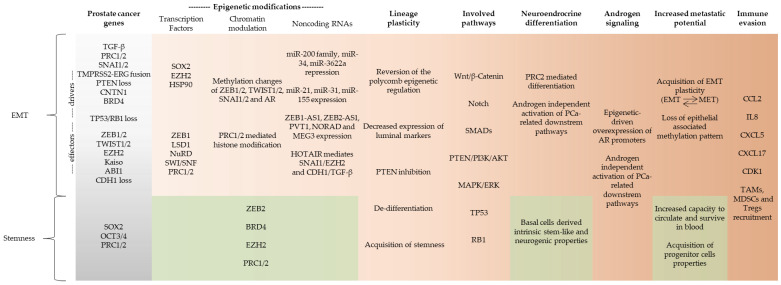
Schematic depiction of regulation of EMT and stemness related to PCa progression. Somatic mutations of prostate cancer are thought to lead to molecular changes in transcription factors, chromatin modifiers, and noncoding RNA involved in EMT and stemness. The consequences of the epigenetic changes initiated by EMT driver transcription factors reprograms chromatin, conferring epigenetic and transcriptional changes to downstream EMT effector genes [16]. These alterations can contribute to plasticity and alterations to other genes and pathways that promote the phenotypic and molecular changes of EMT and stemness during PCa progression [18,62]. The horizontal left to right axis shows the various genetic and epigenetic processes activating pathways of progression (with increasing orange intensity) to mCRPC. Stemness and EMT have closely related prostate cancer somatic genetics (shown as a grey gradient, left side), and specific stemness features during progression are depicted in pale green (lower panels).

**Table 1 genes-12-01900-t001:** Concept definitions.

Castration-resistant prostate cancer (CRPC)
CRPC is an incurable advanced prostate cancer that underwent genomic and phenotypical changes that promoted resistance to androgen deprivation therapy. CRPC overcomes the standard of care therapy and, therefore, its progression is independent of androgen signaling. It is believed that CRPC became resistant to hormonal therapy by reactivation of prostate developmental stem-cell-like gene expression.
Neuroendocrine differentiation
Neuroendocrine differentiation in prostate cancer is a well-recognized phenotypic change by which prostate cancer cells transdifferentiate into neuroendocrine-like cells. NE-like cells lack the expression of AR and prostate-specific antigen (PSA), and are resistant to androgen deprivation therapy.
Double-negative prostate cancer (DNPC)
DNPC are tumors that lack both AR expression and neuroendocrine differentiation.
Epithelial–mesenchymal transition (EMT)
EMT is a latent embryonic programming whereby epithelial cells change from cuboidal- to spindle-shaped morphology, losing their epithelial phenotype to acquire mesenchymal features. It is a pivotal process during homeostasis, but in the context of cancer, EMT programming can be reactivated to promote tumor initiation, invasion, migration, metastasis, and stemness. In this context, EMT drivers are responsible to start the EMT process that promotes expression of downstream EMT effectors.
Mesenchymal–epithelial transition (MET)
MET is the reverse of EMT programming, by which the cells undergo changes to lose the mesenchymal phenotype and acquire epithelial features. MET and EMT work together to promote lineage plasticity.
Lineage plasticity
Plasticity endows cancer cells with the capacity to shift dynamically between a differentiated state, with limited tumorigenic potential, and an undifferentiated or cancer stem-cell-like (CSC) state, which is responsible for long-term tumor growth. During lineage plasticity, cells constantly shift between EMT and MET to acquire therapy resistance and enhance stemness and metastatic potential.
Stemness
Stemness is a state of pluripotency that combines the ability of a cell to perpetuate its lineage, to give rise to differentiated cells, and to interact with its environment to maintain a balance between quiescence, proliferation, and regeneration.
Cancer stem cells (CSCs)
The cells in human tumors, such as PCa, are organized hierarchically, and CSCs comprise a tiny subset of cancer cells that are endowed with tumor-initiating and long-term tumor-propagating capabilities. These tumor-initiating cells display phenotypic and functional features characteristic of normal prostate stem cells and are involved in tumor initiation, metastasis, and drug resistance.
Chromatin remodeling
Chromatin remodeling is the main form of epigenetic control of expression. Through processes such as DNA methylation and histone modification, the chromatin can be opened or closed. A more opened euchromatin is less condensed and favors gene transcription, while a closer and more condensed heterochromatin suppresses gene expression.
Histone modification
Histone modification is a complex epigenetic process by which histone tails are acetylated, methylated, phosphorylated, ubiquitinated, or sumoylated to directly or indirectly alter their affinity to DNA. Histones structure the chromatin, and therefore, the more affinity they have to DNA, the more condensed the chromatin is.
DNA methylation
DNA methylation is a reversible but stable epigenetic process by which a methyl group is attached to the 5-carbon of a cytosine in a CpG dinucleotide, catalyzed by DNA methyltransferase (DNMT). This process allows ligation of methyl-CpG binding-domain proteins (MBDs) that recruit histone-modifying enzymes to promote heterochromatinization. CpG islands, regions rich in CpG dinucleotides, are mostly found in regulatory gene loci, especially promoters and enhancers, and are essential to normal development and cell differentiation.
Noncoding RNA
Noncoding RNAs are single-stranded molecules of RNA that do not encode proteins. The main noncoding RNAs involved in epigenetic processes are micro-RNAs and long noncoding RNAs. Micro-RNAs are small molecules of RNA (19–25 nucleotides) that bind to complementary regions of messenger RNAs (mRNAs) to post-transcriptionally regulate gene expression. Long noncoding RNAs (lncRNAs) comprise RNA species with more than 200 nucleotides that regulate genes expression by controlling nuclear architecture, nuclear transcription, and mRNA stability.

**Table 2 genes-12-01900-t002:** EMT-targeting clinical trials in PCa.

**Biomarkers Investigation**			
**Target**	**Rationale**	**Clinical Trial Identifier**	
Circulating biomarkers of epithelial plasticity	Biomarkers of epithelial plasticity and microtubule interacting protein variants may be related to docetaxel resistance and be enriched in patients failing abiraterone.	NCT02269982	
Circulating tumor cells (CTCs)	Determine whether circulating tumor cells in patients with metastatic progressive castration-resistant prostate cancer or metastatic progressive breast cancer can be captured using a novel mesenchymal marker-based ferrofluid (N-cadherin- or O cadherin-based).	NCT02025413	
CTCs, free DNA, and EMT antigens	Identification of biomarkers that may be predictive of outcome of activity of cabazitaxel treatment in castration-resistant prostate cancer.	NCT03381326	
**Therapeutical Agents**			
**Target**	**Rationale**	**Drug**	**Clinical trials identifier**
AMP-Kinase	Recent evidence shows that the drug may circumvent tumor growth and resistance to castration therapy.	Metformin	NCT01620593 NCT02176161
WEE 1 Inhibitor	Specifically inhibiting WEE1 may restore CDK1 expression and re-establish immuno-mediated attack of mesenchymal-like cells.	Adavosertib	NCT03385655 NCT02465060
Deacethylase inhibitors	Specifically blocking HDACs may prevent histone modifications implicated in cancer.	Romidepsin	NCT00106418 NCT00106301
		Panobinostat	NCT00667862 NCT00878436
		Pracinostat	NCT01075308
		Vorinostat	NCT00330161 NCT00589472
		Phenylbutyrate	NCT00006019
EZH2	Restraining EZH2 may repress PRC2-mediated EMT.	Tazemetostat	NCT04179864
		CPI-1205	NCT03480646
DNMTs	Specifically argeting DNMTs may avoid hypermetilation of CDH1 and tumor suppressors.	Azacitidine	NCT03572387 NCT00384839
		Decitabine	NCT02649790 NCT03572387

Clinical trials that target epithelial–mesenchymal transition and their identifiers from ClinicalTrials.gov. CTCs—circulating tumor cells, WEE1—WEE1 G2 checkpoint kinase, CDK1—cyclin-dependent kinase 1, EZH2—enhancer of zeste 2 polycomb repressive complex 2 subunit, DNMTs—DNA methyltransferases.

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
