# Peer review of "Epithelial–Mesenchymal Transition Signaling and Prostate Cancer Stem Cells: Emerging Biomarkers and Opportunities for Precision Therapeutics"

_genes, 2021, doi:10.3390/genes12121900_

Round 1

Reviewer 1 Report

In this manuscript, the authors provide a detailed discussion on the roles of epithelial-mesenchymal transition (EMT) in prostate cancer. The authors clearly introduce the relationship between somatic mutations, epigenetic regulation, cancer stemness, immune microenvironment, non-coding RNA regulation, and EMT of prostate cancer, which can help to better understand how these essential molecules contribute to cell homeostasis and disease. The manuscript is well written and the illustration is presented in a good quality. This will provide interesting information for the reader of the journal. However, there are still some grammatical and syntax errors in the article. So I think the manuscript can be accepted after grammar and language check.

Author Response

We are grateful for these comments. It was our intention to summarize and update how EMT is intrinsically related to prostate cancer molecular behavior and how that could be of benefit in the clinical settings. We are glad to know that the message was passed clearly. We submitted the manuscript to a meticulous language check and have corrected any grammatical or syntax errors.

Reviewer 2 Report

This manuscript is an informative read on prostate cancer progression driven by EMT and cancer stem cells. It also used a definition table effectively to explain some core concepts and help the audience understand the content better. I suggest the following changes:

  1. The use of terminology such as “EMT driver genes”, “EMT regulators”, and “EMT effectors”, etc. can be confusing. These terminology sound similar but do have different meanings in the context of the article. I suggest consolidating the terminology and eliminating redundant ones. It would also be very helpful to clarify what exactly each terminology means, such as what makes a factor an EMT driver vs. an effector, in the beginning of the article.
  2. The authors can simply use the citation number instead of “reviewed in…”. For example, “reviewed in 15” in line 84 can be simplified to “15.
  3. In section 2, the authors wrote “There is a strong association between the expression of the E-cadherin/catenin complex and progression in various types of carcinomas, including PCa.” Please be more specific about what this association is. The sentence makes it sound like there is a positive correlation between E-cadherin expression and PCa progression, while earlier in the paragraph it describes how EMT and progression of PCa are associated with loss of E-cadherin. It could be interpreted as conflicting messages, so more clarification is neccesary.
  4. Given “emerging biomarkers and opportunities for precision therapeutics” is part of the title, I suggest expanding on the topic in the manuscript. Specifically, when discussing a biomarker, the authors can indicate whether it is prognostic vs. predicative. The authors did so in some places but not all. The discussion on precision medicine opportunities is also quite limited. I recommend the authors propose some predicative biomarkers to stratify patient populations for the purpose of precision medicine.
  5. Once an acronym is established with the full name in the beginning of the article, the acronym can be used consistently throughout the article and authors no longer need to spell out the full names. For example, the full name of EMT is unnecessary in line 84 and 89.
  6. There are some grammatical errors, which need to be corrected. For example, there should be an “in” after “increase” in “an increase cell migration and invasiveness due to activation …” in line 95. I suggest thorough editing of the manuscript by professional life sciences editors.

Author Response

Response 1: This is a very helpful suggestion as the distinction between EMT regulation, drivers and effectors could be confusing for readers. EMT regulator was used to generally address any process, gene or pathway that could modulate EMT programming; EMT driver was used to refer to genes and pathways that initiate EMT programming; EMT effector was used to allude to downstream genes and pathways that effectively promote the mesenchymal-like phenotype. We clarified this point in the text on page 3 (first paragraph, TABLE 1) and avoided redundant terms as suggested. 

Response 2: “reviewed in” was used to address when a strong review in the area was cited. We simplified that, as suggested.

Response 3: We agree with the reviewer that the original text appears to be conflicting with the prior mention of e-cadherin. As suggested, we now provide more details about the nature of the association in the revision of the manuscript (lines 109-111).

Response 4: We highlighted more potential predictive biomarkers to be taken advantage of in the precision therapeutics settings (lines 173-174, 230-232, 327-329, and 538-540).

Response 5: The manuscript was reviewed to remove full spelling when an acronym could be used.

Response 6: We submitted the manuscript to a meticulous language check.

Reviewer 3 Report

This is an outstanding and very informative review on the molecular mechanisms of EMT as they relate to progression of prostate malignancies.   It is very easy to follow the logic of all the different sections of this manuscript.  Well done.

Author Response

We thank the reviewer for supporting and appreciating our review.